# Comparison of Structural Integrated Piezoceramics, Piezoelectric Patches and Strain Gauges for Condition Monitoring

**DOI:** 10.3390/s22228847

**Published:** 2022-11-16

**Authors:** Jonas Maximilian Werner, Max Engelmann, Marek Schmidt, Christian Titsch, Martin Dix, Welf-Guntram Drossel

**Affiliations:** 1Professorship for Adaptronics and Lightweight Design in Production, Chemnitz University of Technology, Reichenhainer Straße 70, 09126 Chemnitz, Germany; 2Professorship Production Systems and Processes, Chemnitz University of Technology, Reichenhainer Straße 70, 09126 Chemnitz, Germany; 3Fraunhofer Institute for Machine Tools and Forming Technology IWU, 09126 Chemnitz, Germany

**Keywords:** condition monitoring, piezoelectric sensors, strain gauges

## Abstract

This paper presents a new approach to the structural integration of piezoceramics into thin-walled steel components for condition-monitoring applications. The procedure for integrating the sensors into metal components is described, and their functionality is experimentally examined with a 2 mm-thick steel sheet. The signal quality of the produced sensors is investigated in a frequency range from 100 Hz to 50,000 Hz and is compared with the results of piezo patches and strain gauges under the same conditions. The results show that due to a higher signal-to-noise ratio and a better coherence, the structurally integrated piezoceramics and the piezo patches are more qualified sensors for vibration measurement in the examined frequency range than the strain gauges. The measurements also indicate that the patches provide higher amplitudes for the frequency range up to 20 kHz. Beyond that, up to 40 kHz, the integrated sensors supplied higher amplitudes. The better signal quality in different frequency ranges as well as the different manufacturing and application methods can be interpreted as an advantage or disadvantage depending on the boundary conditions of the condition-monitoring system. In summary, structural integrated piezoceramics extend the options of monitoring technology.

## 1. Introduction

Industrial production and manufacturing processes are subject to high-quality requirements. In order to meet these requirements, machines have to be monitored, errors have to be detected at an early stage, and machine failures have to be avoided [1,2]. Consequently, concepts such as condition-based predictive maintenance are increasingly moving into focus. Condition-monitored processes enable the real-time measurement of machine and operating states. Based on the collected data and a corresponding evaluation, it is possible to obtain, for example, the following benefits [1,2,3,4]:Increase in process safetyEarly detection of system deviationsEasier identification of error causesFaster reactions in case of possible machine downtimeReduction in energy costs, maintenance and repair costsPrognosis of remaining machine service life and wear of partsImproved occupational safety for employeesOptimization of the quality of the process sequences and products

The need for condition-monitoring systems can be found in a wide range of industries. They are used to ensure process reliability. Such systems are particularly necessary for continuously running systems such as wind turbines as well as components with increased wear, such as milling tools [2,5,6].

Depending on the requirements and boundary conditions, various sensors are suitable for condition monitoring. The functions of these sensors are usually based on electrodynamic, piezoelectric, eddy current, inductive, magnetic and thermal technologies [3]. Especially when high measurement accuracy is required for dynamic processes, piezo sensors are the preferred choice. Compared with other types of sensors, piezoelectric sensors are characterized in particular by properties such as high amplitude over a wide frequency range, fast response time and a high modulus of elasticity [7]. Furthermore, piezoelectric sensors have a simple structure, high resolution and require minimal installation space [7,8].

Piezo sensors have another significant advantage over other sensor types. They work either as a sensor or as an actuator. Thus, they are particularly suitable for Lamb wave based structural health monitoring (SHM) of metallic structures [9,10]. Lamb waves can be used especially for monitoring large metal parts due to their ability to travel over long distances. Therefore, piezo-based SHM technology is particularly suitable for monitoring aircraft parts such as wings [11,12]. Piezo transducers are applied in an array on the surface of the aircraft wings to track crack propagation. Piezo elements, which have a laminar structure and are embedded between elastic foils like piezo patches [13], macro fiber composites (MFC) [14] or smart layer sensors [15,16,17], are often used. Generally, these piezoelectric transducers are applied to the surface by means of adhesives and are then electrically contacted. The pre-packaged transducers offer advantages over monolithic piezoelectric elements. Due to the embedding, the transducers are completely electrically isolated from the component to be measured and offer simple electrical contacting via solder pads [13]. Furthermore, the design of the transducer allows a more flexible application to curved surfaces.

However, there are also disadvantages, such as the possible miniaturization of the transducers, which is limited by the pre-packing. The material used for embedding also determines the limits of the application temperatures, which usually end in the range of 150 °C [18]. The Curie temperature limits the functionality of the piezoceramic material and can be up to 350 °C for specific piezoceramics [19].

Further, debonding is a general problem in terms of long-term stability [20]. Moreover, a damping of the transmission signal is to be expected due to the adhesive and embedding layers [21]. This leads to a deterioration in the dynamic behavior, which has already been studied in various scientific papers. For example, the deviations between simulated vibration systems and experimental values in the use of piezoelectric fans [22] as well as in elastic wave extinction and detection [23,24] have been explained by the applied adhesive layers. Furthermore, the effects of adhesive layers on wave propagation were specifically investigated in [25,26]. A significant influence of the adhesive layer on the resulting contact shear stresses was found. A thicker adhesive layer leads to a damping or lag effect on the load transfer between the lead zirconium titanate (PZT) element and the host structure.

Limited scientific papers have been published focusing on the avoidance of adhesive layers and associated disadvantages for piezoelectric transducers in thin-walled metal structures [27,28,29]. One of these, an approach for piezoelectric elements with a round cross-section, is described in [27] for the fabrication of a vibration viscosity sensor. Stainless steel wires are pressed into the surface of an aluminum sheet together with a copper foil. Subsequently, the steel wires are removed, and piezoelectric elements with a round cross-section and a metal core are inserted into the cavities. Finally, a metal-core piezoelectric fiber/aluminum composite is fabricated via a second aluminum sheet and the so-called interphase forming/bonding (IF/B) method (Figure 1a).

Another approach is the structural integration of piezoceramic elements with rectangular cross-sections into aluminum sheets by joining and by forming [28,29]. The piezoceramic elements are inserted in the cavities of a prefabricated microstructure in the aluminum sheet. Then, a punch deforms the aluminum material so that the piezoceramic elements are clamped in the structure (Figure 1b). In this way, a direct electromechanical coupling between the joining parts is achieved without adhesive layers.

In this study, the last-mentioned integration process is modified in order to realize a structural integration of monolithic piezoceramic elements into steel components. The challenge is to achieve a direct acoustic coupling with a sufficiently high preload of the piezoceramic elements without damaging them. Advantages are expected in terms of higher operating temperatures and better transmission behavior due to the low damping effect. The method of structural integration also offers the possibility of integrating the sensors into the manufacturing process of components and assemblies at an early stage. For instance, the piezo elements can already be structurally integrated into a semi-finished product by means of forming technology. After this, the semi-finished products can be shaped by deep drawing, while the sensors retain their functionality [30]. This allows sensors to be applied to locations on the final part that cannot be reached at a later stage of manufacturing. Furthermore, for future applications the structural integrated piezoceramics (SIP) might be sealed to create an even surface after manufacturing as opposed to other sensor types.

The intention is to expand the field of applications for piezoelectric sensors integrated by forming technology and to provide alternative monitoring sensors to the piezo patch solutions. For this purpose, the quality of the sensor signal of the pressed-in sensors is examined and compared with the sensor signal of bonded piezo patches in experimental tests. In addition, strain gauges (SG) are applied to the steel components. Strain gauges are widely used in industry for condition-monitoring applications, are easy to apply and are very cost-effective [31,32]. Upper frequency limitations are not given by the manufacturer. Therefore, the suitability of strain gauges for the measurement of vibrations in the ultrasonic frequency range has already been evaluated [33]. Advantages of the strain gauges are their low weight and, therefore, a negligible influence on the dynamic behavior of the system. Additionally, the measured signal shows a good signal-to-noise ratio and the application of more than one strain gauge would enable the detection of vibration modes in the observed system. The experiments are intended to provide information on the excitation ranges in which the strain gauges or the piezoelectric sensors are qualitatively more useful, with particular focus on the transfer function. In addition, the level of the measured amplitudes, the noise behavior and the transverse sensitivity are compared between the different sensor types.

In this report, three different sensors are compared regarding the quality of their detected measurement signals. For this, these sensors were applied or integrated into a metal sheet, which was excited with sine waves at different frequencies using a piezoelectric patch as an actuator in the middle of the sheet. For the first time, the SIPs are compared to strain gauges and piezoelectric patches and differences in their signal quality at different frequencies are shown. It is demonstrated that the SIPs exhibit a higher signal-to-noise ratio than the other sensors at higher frequencies and therefore show a better suitability for future use in condition monitoring or SHM systems, for example for damage detection in aircraft wings. This approach of structurally integrated piezoceramics avoids the usage of adhesive layers and aims at a higher acoustic coupling between the piezoceramic and the surrounding host structure.

In the following experiment, the three sensors used, strain gauges, piezoelectric patches as well as SIPs, are described with an emphasis on the manufacturing of the SIPs. Furthermore, the setup for the experiments is shown as well as the necessary preliminary investigations, in this case a numerical analysis of the metal sheet and an experimental analysis of the vibration modes of it using a 3D scanning vibrometer. Additionally, the noise of the sensors was measured, and an experimental plan was designed.

In Section 3, the results of the experiments are evaluated with an emphasis on the signal-to-noise ratio, the orientation of the sensors in relation to the actuator and the signal quality depending on the excitation frequency used.

## 2. Materials and Methods

The different types of sensors strain gauge, piezoelectric patch and the structural integrated piezoceramics (SIP) are compared in experimental tests. Table 1 summarizes the names and manufacturers of the different sensor types. The SIP sensors operate with the longitudinal d33 effect. Therefore, specific piezo patches which also use the piezoelectric d33 effect [18] are selected for comparison. However, the patch can only be applied to surfaces, e.g., by means of an adhesive bond. This also applies to the strain gauge, whose operating principle is based on the detection of the change in electrical resistance. For the comparison, strain gauges with an active element size similar to that of the patch are selected. The chosen strain gauges are conventional strain gauges with a gauge factor of 2.04 and a nominal resistance of 350 Ω. The temperature compensation and more specifically the temperature coefficient of the strain gauges can be neglected since the experiments will be carried out at room temperature and a significant build-up of heat is not expected.

The experimental approach involves a defined excitation of a steel sheet on which the mentioned sensors are applied and an analysis of the resulting sensor signals is carried out. Important sensor characteristics such as sensitivity or noise behavior can thus be evaluated and quantitatively compared.

### 2.1. Structural Integrated Piezoceramics (SIP)

The sensor principle is based on preliminary work with the structural integration of piezoelectric elements with a rectangular cross-section, which is described in Section 1. The main difference is the integration of the piezoelectric elements into stainless steel sheets of X5CrNi18-10 [36], which are used in the aerospace industry, instead of the aluminum alloy EN AW-5083 [28,29]. It follows that greater forming forces are required, because the yield strength increases from 125 MPa to 210 MPa [37,38]. However, greater forming forces lead to a greater likelihood of fracture of the brittle ceramics. Therefore, a new approach is developed for integration into steel components. As shown in Figure 2, a ceramic insulator and two piezo fibers, each with a rectangular cross-section, and a copper wire with a square crosssection, are inserted into the cavity of a steel component with a defined position. An assembly gap of 0.08 mm between the components enables easy assembly by hand. Subsequently, a flat die applies a forming load to the exposed surface of the copper wire via a hydraulic machine vise (Hilma KNC 100 by STARK Spannsysteme, Rankwell, Austria). Due to the resulting plastic deformation of the copper, the piezo fibers are force-locked into the cavity and preloaded. A particular challenge is the selection of the optimal forming load, since a too high preload force F_pre_ could destroy the piezoceramic fibers. However, if the punch force is too low, there is the risk of an unstable frictional connection. According to initial experience with the integration method, the optimum forming load for the sensor setup described above is P_form_ = 265 N/mm^2^. An optical inspection using a microscope shows no, or only a few, small visible cracks in the piezoceramics and the connection was not detachable with a bare hand.

The structural integrated piezoceramic fibers are previously made from polarized piezoceramic plates (thickness of 0.26 mm) with the manufacturer’s designation “M1100” from Johnson Matthey [35] by a parting-off grinding process using a peripheral wafer dicing saw (Logitech APD1, Glasgow, UK). For this purpose, lead zirconate titanate (PZT) plates are bonded to a glass slide with a heat-soluble wax, cut with the peripheral wafer dicing saw, and then debonded from the glass slide. The PZT plates are coated with a gold layer on the two large surfaces, which act as electrodes on the contact surfaces between metal elements and the piezoceramic. The dimensions of the fibers are specified in Figure 2. The cavity of the steel component required for integration is made by end milling. The ceramic insulator (Al_2_O_3_) is also manufactured by a grinding process with the peripheral wafer dicing saw. The width of the ceramic insulator is smaller than that of the cavity for two reasons. Firstly, there must be a clearance for insertion. Secondly, radii remain on the inner edges of the cavity due to the manufacturing process so that the bottom of the cavity is not flat over the full width. The reduced width of the ceramic insulator is therefore necessary to ensure its full area contact on the bottom and thus ensure a defined position. The advantage of the ceramic insulator compared to the coated insulator as in [39] is a lower manufacturing cost.

Figure 3 illustrates the cross-section of a successfully fabricated SIP in a steel sheet and its functional principle, which is based on the use of the longitudinal piezoelectric d33 effect. The structurally integrated piezoceramic fibers are arranged in an electrical parallel connection. Thus, the steel component is the ground electrode, while the formed copper wire acts as the counter-electrode at which the signal is generated. The ceramic insulator at the bottom of the cavity prevents short circuits between the electrodes. Vibrations and resulting elastic deformations of the steel component are transmitted directly to the piezo fibers through the contact surfaces and lead to mechanical tension as well as strain. This, in combination with the d33 effect, causes a change in voltage between the electrodes. By measuring this voltage change, the oscillation state of the steel component can finally be evaluated.

### 2.2. Experimental Setup

The sensors specified in Table 1 are applied, as shown schematically in Figure 4, to two metal sheets measuring 200 mm × 200 mm × 2 mm with an even distance between them.

The sheets are made of the material X5CrNi18-10. A piezoelectric patch is applied in the center, which acts as an actuator for targeted excitation of the sheet. The connections between the steel sheet and the piezoelectric patches as well as the strain gauges are made by using an adhesive bond. As described in Section 2.1, the piezo patches were force-fitted into previously milled cavities. The uniform arrangement of the sensors around the actuator ensures a comparable excitation of the sensors.

The entire test setup, as well as the used measurement equipment, is shown in Figure 5. The metal sheets are clamped separately at the corners into an aluminum frame via a force-fit connection. The operating principle of strain gauges is based on the Wheatstone bridge circuit. This means the change in electrical resistance of the resistors located in the bridge circuit is proportional to the strain and can be determined by measuring the electrical output voltage of the bridge. Thus, a voltage supply is required to feed the bridge circuit. The voltage supply is established by the universal measurement amplifier QuantumX MX410B from HBM, which also detects the output voltage of the strain gauges and transmits the analog measurement signal to a PC with an integrated analog-to-digital converter. The piezoelectric patches used for sensing as well as the SIP do not require an external power supply and are directly connected to the analog-to-digital converter of the PC. A signal generator controls the actuator, with the consequence that different excitation modes and amplitudes are possible. In addition to the applied sensors, the 3D scanning vibrometer PSV-500 from the manufacturer Polytec GmbH (Waldbronn, Germany) records the induced vibration. Thus, reference values and occurring vibration modes can be determined and taken into account. In future applications, the signals can also be measured using other measurement systems, where real-time evaluations are possible. The 3D scanning vibrometer was used to measure all sensor signals as well as the optical measurement with one single measurement system.

### 2.3. Preliminary Investigations

The comparison of the applied sensors is based on the contrast of measured signals at the same excitation in each case. Since the respective excitation of the sensors depends on the control and the characteristics of the actuator as well as on the resulting dynamic transmission behavior of the sheet metal, preliminary investigations are necessary to find suitable excitation frequencies at which the vibration is identical in all four sensors.

First, the natural frequencies and vibration modes of the clamped sheet metal were determined with the aid of a numerical modal analysis using the ANSYS software as a preliminary investigation to obtain an estimate for the expected vibration shapes and to compare the calculated vibration shapes to the measured ones later on. The simulation model is shown in Figure 6. The geometry of the steel sheet was modeled according to the dimensions in Figure 4 and consisted of about 55,000 tetrahedral elements of the element type SOLID187. Its properties are described by the material-related modulus of elasticity of 200 kN/mm^2^ and a density of 7.9 kg/dm^3^ [40]. To take the clamping of the sheet into account, fixed supports were defined on the surfaces in the corner areas.

By analyzing the calculated natural vibration modes up to 50 kHz, it can be seen that the vibration responses of the sheet metal respective to the amplitudes are almost the same in all sensor positions and thus an identical excitation of the sensors can be estimated. An example of a calculated vibration mode and the identical excitation of the sensors is shown in Figure 6. The simulation results are as expected due to the ideal symmetry of the simulation model and the symmetrical arrangement of the sensors.

The upper limit of 50 kHz was chosen because of the limitations of the used amplifier. While measuring four sensor signals, the highest sampling rate possible is 96 kHz. Thus, according to the Nyquist–Shannon sampling theorem the highest frequency that can be measured correctly is lower than 48 kHz.

To verify the assumption of equal excitations derived from the simulation, the vibration shapes of the metal sheet were examined experimentally on the test rig shown in Figure 5. For this purpose, the vibration behavior was measured at 12 points on the top of the frame using the 3D laser vibrometer. The sheet was excited with a swept sine wave from 100 Hz to 50,000 Hz. The upper limitation for the frequency is caused by the maximum sampling rate of the amplifier used for the strain gauges. The measured resonance frequencies fit with the results from the simulation. The vibration shapes are mainly out-of-plane vibrations with nearly no displacements in the x and y dimensions. The only noteworthy in-plane vibration is at approximately 11.23 kHz and is shown as an example in Figure 7. It shows the absolute displacement of each measurement point. The highest displacement in the x and y direction at the border of the measurement grid is 9.5 nm and 8.3 nm, respectively. The measurements show no noteworthy differences in the vibration in the sensor locations at the considered frequencies and therefore confirm the simulation results. Thus, for the following experiments it is assumed that an almost identical excitation of the sensors is given when the sheet is excited with actuator frequencies from 0 to 50 kHz.

In Table 2, the measured noise and offsets for each individual sensor are shown. For the strain gauges, the value is given in mV and not µm/m with the result that a comparison between the sensors is feasible. Each of the used sensor systems shows an offset caused by a prestress during the application. These offsets are accounted for in the later analysis. When we compare the noises of the sensor systems, it is clear that the strain gauges produce greatly more noise than the SIP and the patches. This may indicate a higher signal-to-noise-ratio for the SIP and the patches, but this needs to be evaluated during the experiments. The reason for the higher noise of the strain gauges may be the amplifier and the voltage supply.

### 2.4. Experimental Plan

For the experiments, specific frequencies were chosen to evaluate the sensor performance on a broad frequency spectrum. To reduce the scope of the experiments, the frequency steps for the excitation with a fixed frequency were bigger for higher frequencies, as shown in Table 3. For lower frequencies, the experiments were carried out at 100 Hz, 300 Hz and 500 Hz.

## 3. Results and Discussion

When we compare the different sensor signals in the time domain, as seen in Table 4, it is apparent that the SIP and the patch have a higher signal-to-noise ratio than the strain gauges. For the comparisons, only sensors with the same alignment relative to the actuator, as listed in Table 2, were evaluated. Table 4 shows the results for the noise of the sensors as well as for an excitation with different frequencies. The chosen frequencies were not identified as vibration modes of the steel sheet and the distribution of the vibrations at the sensor points is identical.

The strain gauge shows a relatively high noise in comparison to the SIP as well as the patch. This noise was observed in previous works, carried out by the authors, too [33]. In comparison to the measured signal, the strain gauge exhibits a very low signal-to-noise ratio. For an excitation with a frequency of 20 kHz, the maximum values for the noise and the measured signal are 8 mV and 12 mV, respectively. The SIP and the patch have a much higher signal-to-noise ratio. For the patch, the values are 0.5 mV for the noise and 8 mV for the measured signal and for the SIP the maximum amplitudes are 0.5 mV and 30 mV. For the parallel alignment of the sensors, the results are similar and shown in Table 4. The higher values for the parallel alignment are expected since all of the sensors have a lower transverse sensitivity. The high signal-to-noise ratio indicates a suitability of the patch as well as the SIP for the measurement of vibrations with low amplitudes. Some values for the strain gauges are not listed in Table 4 because the measurement showed no discernible difference between the noise and the measured signal. The noise and the measured signal of the sensors are shown in Figure 8. Additionally, the signal-to-noise ratios of the sensors for different frequencies are shown in Table 5.

Regarding the coherence of the excitation signal and the measured signals, the strain gauges show the lowest coherence of all the sensor signals, as seen in Figure 9. This was to be expected from the signals in the time domain. The strain gauges exhibit a high noise while measuring comparatively low strains. Strain gauge 2 from sheet 1 shows even lower coherence than the other strain gauges. The reason for this is the perpendicular alignment of SG 2 in relation to the excitation signal. The piezo patches as well as the SIP show a very high coherence of nearly one over the whole frequency spectrum, which demonstrates a high relation between the excitation signal and the measured signal and therefore a better signal transmission. It does not appear that the alignment of the sensors has an influence on the coherence.

## 4. Conclusions

The paper presented a method for the structural integration of piezoceramic elements in steel sheets for sensory applications. The functionality and signal quality were investigated in a frequency range from 100 Hz to 50,000 Hz and were compared with commercially available piezo patches and strain gauges. The suitability of the SIPs as vibration sensors and thus as condition-monitoring sensors was proven in the experimental tests. As is common with the use of sensors, all boundary conditions for the structure or component to be monitored must be considered and the appropriate sensors selected specifically for the application. Thus, the outcomes of the investigations also show the application-specific advantages or disadvantages of the tested sensors.

The results show that the SIPs and the piezo patches indicate a higher potential for measuring vibrations than the strain gauges, which was expected due to the high sensitivity of the piezo material. They exhibit a much higher signal-to-noise ratio than the strain gauges and a better coherence in the considered frequency range. Strain gauges seem to be inadequate for the investigated frequency range due to the high noise ratio. This could not be deduced directly from the manufacturer’s specifications, but it is in line with the preferred application of strain gauges for predominantly static measurements. Furthermore, both the SIP and the patches exhibit a higher transverse sensitivity. The transverse sensitivity of the strain gauges is very low. Depending on the use case, this may be beneficial if the direction of vibration is of importance. Additionally, the SIPs show a higher transverse sensitivity than the patches, especially at higher frequencies.

As seen in Table 4, the SIPs and the patches may be advantageous for different frequency ranges. While the patches have higher amplitudes than the SIPs up to 20 kHz, this changes at higher frequencies, so the SIPs may be advantageous. This is the case for the perpendicular as well as the parallel alignment of the sensors. However, statements about possible damping effects due to the adhesive layer could not be made from this principal test.

Due to the same underlying operating principle for SIPs and piezo patches, the following section summarizes further advantages and disadvantages of these two sensor types. The operating temperatures of the SIPs are determined by the Curie temperature of the piezo material and can therefore be higher than those of the piezo patches. In the latter case, the surrounding insulation material or the used adhesive limits the application temperature. In addition, it is expected that a higher degree of miniaturization of SIPs can be achieved in contrast to piezo patches, due to the elimination of the encapsulating material. During application, the piezo patches are beneficial with their uncomplicated and fast assembly as well as simple electrical contacting via solder pads. The insertion of an integration cavity for the SIPs and the press-fit process mean that the application effort is more extensive. Furthermore, this is a permanent modification of the component. This structural integration process is advantageous in that the sensor disappears completely into the surface and, for example, the cavity can be used for cable routing and could be closed again with epoxy resin. Piezo patches usually result in a surface protrusion of the component. In the case of the long-term use of piezo patches, delamination, adhesive aging and the associated effects on sensory behavior must be taken into account. Therefore, depending on the desired frequencies the sensor should be able to measure and the conditions of the application, the SIPs may be an improvement on other currently used sensors.

Effects and changes occurring during the long-term use of SIPs, such as changes in the press-fit connection, will be investigated in future studies. Finally, sensors based on structural integrated piezoceramics expand the possible configuration options for condition-monitoring systems.

## Figures and Tables

**Figure 1 sensors-22-08847-f001:**
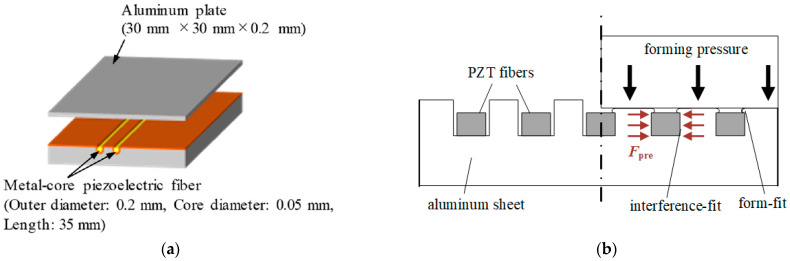
Concepts for the application of piezoelectric transducers without adhesive layers; (**a**) Interphase forming/bonding of piezoelectric elements with round cross-section and metal core [27]; (**b**) Integration of piezoelectric elements with rectangular cross-section by joining by forming [29].

**Figure 2 sensors-22-08847-f002:**
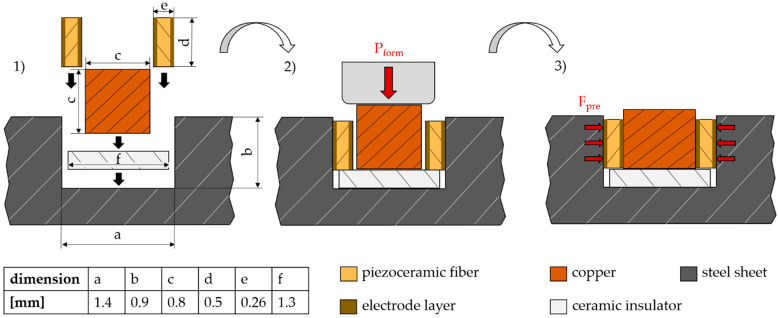
Components and steps for manufacturing the SIP; (**1**) Assembly, (**2**) Joining by forming, (**3**) Joined components.

**Figure 3 sensors-22-08847-f003:**
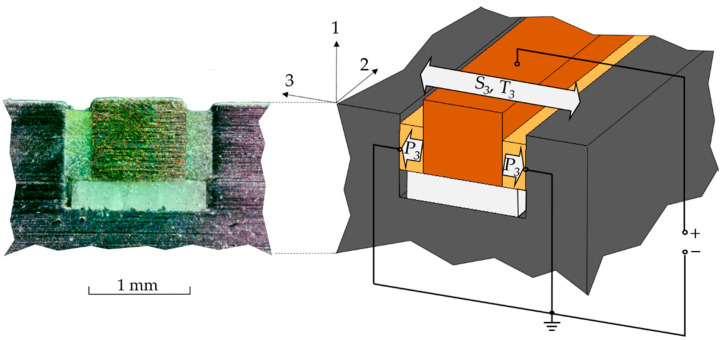
Cross-section (**left**) and functional principle of the SIP (**right**); P_3_: polarization, S_3_: strain, T_3_: tension.

**Figure 4 sensors-22-08847-f004:**
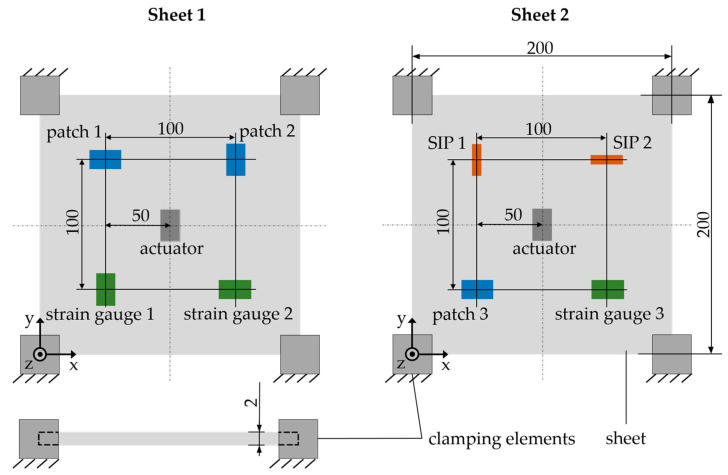
Sensor and actuator arrangement on the sheets (dimensions in millimeters).

**Figure 5 sensors-22-08847-f005:**
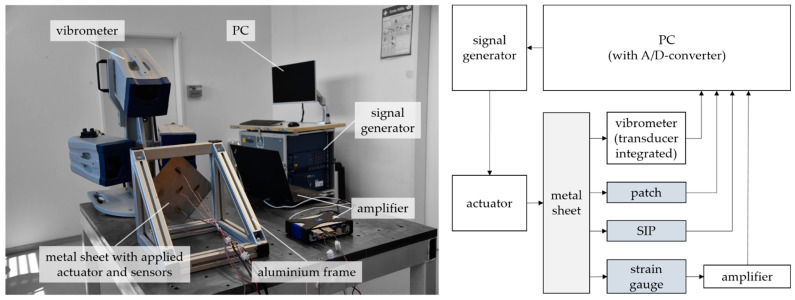
Experimental setup.

**Figure 6 sensors-22-08847-f006:**
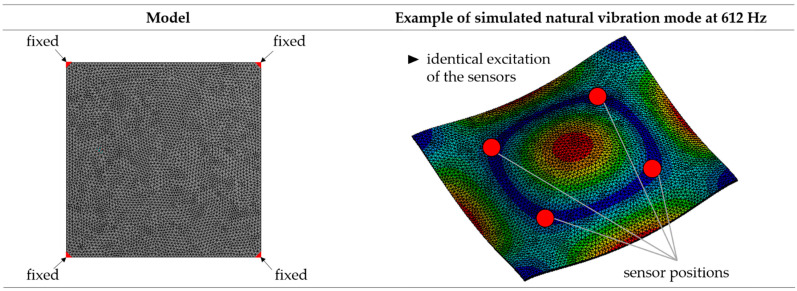
Simulation model for modal analysis of the clamped sheet (**left**) and example of calculated natural vibration mode (**right**).

**Figure 7 sensors-22-08847-f007:**
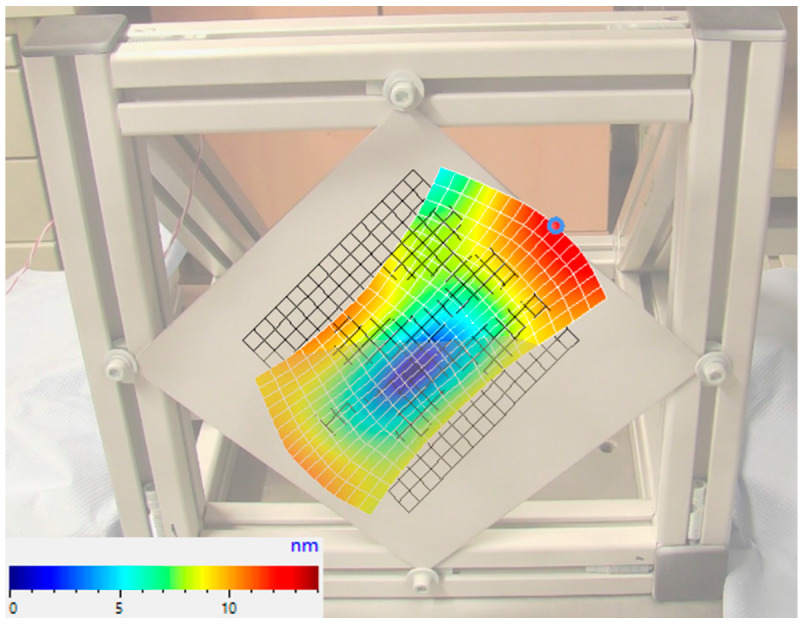
In-plane vibration mode at 11.23 kHz.

**Figure 8 sensors-22-08847-f008:**
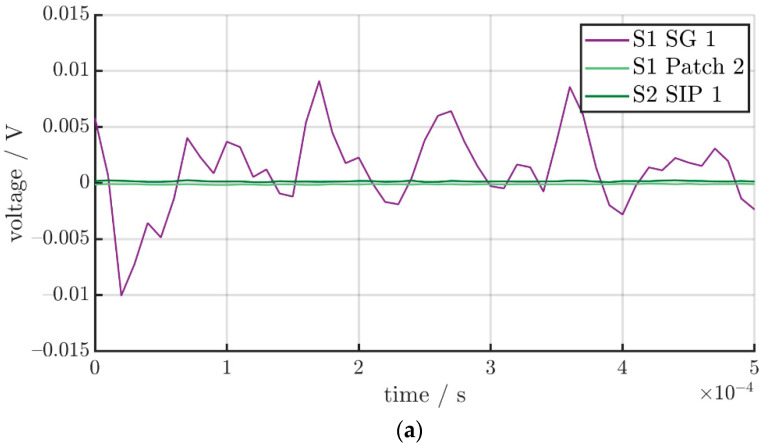
Noise and measured signals of: (**a**) Noise of all sensors with a parallel alignment in relation to the actuator; (**b**) Measurement signal of all sensors with a parallel alignment at 20 kHz (**c**) Noise of the sensors with a perpendicular alignment (**d**) Measurement signal of all sensors with a perpendicular alignment at 20 kHz.

**Figure 9 sensors-22-08847-f009:**
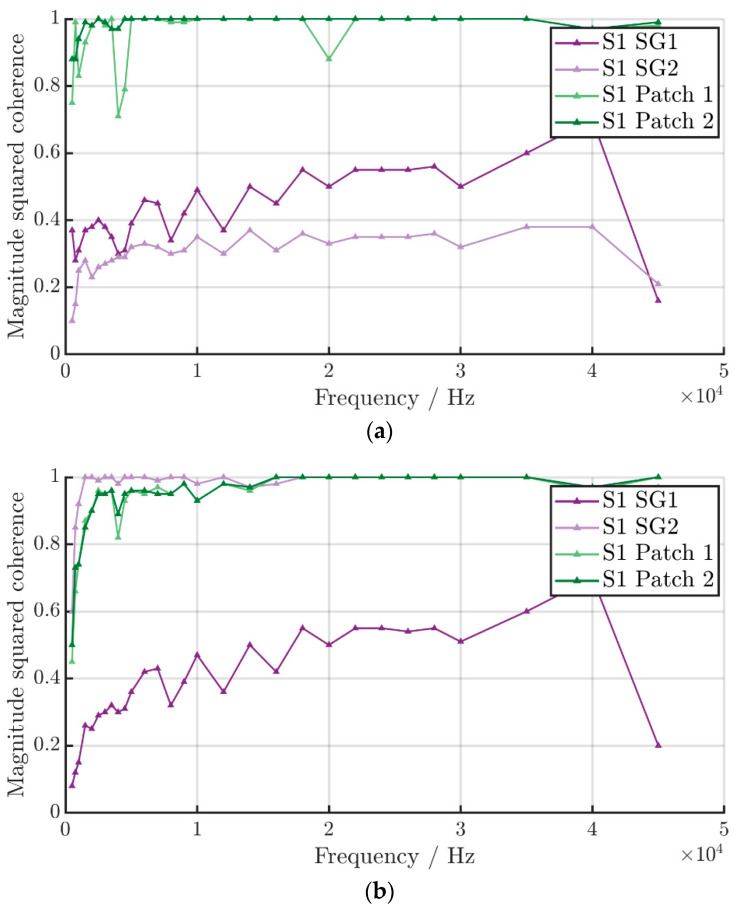
Coherence of the sensor signals of (**a**) Sheet 1 and (**b**) Sheet 2.

**Table 1 sensors-22-08847-t001:** Overview of used sensor types.

	Strain Gauge [34]	Piezoelectric Patch [18]	Structural Integrated Piezoceramics (SIP)
Name	Type: LY4-7-09-350-3-2	P-878 DuraAct Power Patch Transducer	SIP
Manufacturer	Hottinger Baldwin Messtechnik GmbH (HBM), Darmstadt, Germany	PI Ceramic GmbH, Lederhose, Germany	Raw Material from Johnson Matthey, London, UK [35]
Sensor effect	Electrical resistance	Piezoelectric d33-effect	Piezoelectric d33-effect
Size of active element	10 mm × 5 mm × 0.081 mm	15 mm × 5.4 mm × 0.6 mm	5 mm × 1.4 mm × 0.26 mm
Piezoelectric charge coefficient [10–12 C/N]	/	640	400
Transversal contraction	−0.2	~−0.3	~−0.3

**Table 2 sensors-22-08847-t002:** Offsets and noise of the individual sensors.

Sensor	Noise	Offset	Alignment
Sheet 1 Patch 1	±0.5 mV	−70 mV	perpendicular
Sheet 1 Patch 2	±0.5 mV	−19 mV	parallel
Sheet 1 SG 1	±8 mV	+12 mV	parallel
Sheet 1 SG 2	±8 mV	+11 mV	perpendicular
Sheet 2 SIP 1	±0.5 mV	−70 mV	parallel
Sheet 2 SIP 2	±0.5 mV	−20 mV	perpendicular
Sheet 2 Patch 3	±0.5 mV	−17 mV	perpendicular
Sheet 2 SG 3	±11 mV	−32 mV	perpendicular

**Table 3 sensors-22-08847-t003:** Experimental Plan.

Frequency Range	Frequency Step Size
500–1000 Hz	250 Hz
1000–5000 Hz	500 Hz
5000–10,000 Hz	1000 Hz
10,000–30,000 Hz	2000 Hz
30,000–50,000 Hz	5000 Hz

**Table 4 sensors-22-08847-t004:** Values of the noise and the measured signal for an excitation with 5, 10, 20 and 40 kHz.

Sensor	Alignment	Noise	Signal 5000 Hz	Signal 10,000 Hz	Signal 20,000 Hz	Signal 40,000 Hz
Sheet 1 Patch 1	perpendicular	0.5 mV	60 mV	54 mV	8 mV	10 mV
Sheet 1 SG 2	perpendicular	8 mV	only noise	only noise	12 mV	only noise
Sheet 2 SIP 2	perpendicular	0.5 mV	31 mV	49 mV	30 mV	33 mV
Sheet 1 Patch 2	parallel	0.5 mV	55 mV	72 mV	52 mV	10 mV
Sheet 1 SG 1	parallel	8 mV	17 mV	37 mV	50 mV	only noise
Sheet 2 SIP 1	parallel	0.5 mV	26 mV	48 mV	48 mV	21 mV

**Table 5 sensors-22-08847-t005:** Signal-to-noise ratios of the sensors.

Sensor	Alignment	Noise	Signal-to-Noise Ratio			
			5000 Hz	10,000 Hz	20,000 Hz	40,000 Hz
Sheet 1 Patch 1	perpendicular	0.5 mV	120	108	16	20
Sheet 1 SG 2	perpendicular	8 mV	only noise	only noise	1.5	only noise
Sheet 2 SIP 2	perpendicular	0.5 mV	62	98	60	66
Sheet 1 Patch 2	parallel	0.5 mV	110	144	104	20
Sheet 1 SG 1	parallel	8 mV	2.13	4.63	6.25	only noise
Sheet 2 SIP 1	parallel	0.5 mV	52	96	96	42

## Data Availability

Not applicable.

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
