# Peer review of "Comparison of Structural Integrated Piezoceramics, Piezoelectric Patches and Strain Gauges for Condition Monitoring"

_sensors, 2022, doi:10.3390/s22228847_

Round 1

Reviewer 1 Report

In Table 1 could also be useful to indicate the thickness of the sensors.

Line 155: Then, a flat die applies a 155 forming load to the exposed surface of the copper wire.

What is the equipment used to apply that load?

The ceramic insulator (Al2O3) is also manufactured by a parting-off grinding process.

Any details about that process? To process the Al2O3 is a complicate task…

It will be interesting to know more details about the simulation model in ANSYS.

A signal-noise ratio in Table 4 could be interesting.

As a general comment, Is there another way to measure the signal displacement than using a vibrometer?. A low-cost circuit, where the frequency output is measure, could be interesting thinking in a final application of the sensor.

Author Response

Dear reviewer,

Thank you for your comments on our manuscript. Please find attached our answers these.

Best regards,

Jonas M. Werner

Reviewer 2 Report

Please find my comments in the attached file. 

Author Response

(The authors gave the same response as above.)

Round 2

Reviewer 2 Report

The authors have addressed my comments adequately. I accept the paper at this stage.